# Comparative mitogenomic analysis provides evolutionary insights into *Formica* (Hymenoptera: Formicidae)

**Min Liu**[1,2,3,4☯], **Shi-Yun Hu**[1,2,3,4☯], **Min Li**[1,2,4], **Hao Sun**[1,2,3,4], **Ming-Long Yuan**[1,2,3,4]*

**1** State Key Laboratory of Herbage Improvement and Grassland Agro-ecosystems, Lanzhou University, Lanzhou, Gansu, China, **2** Key Laboratory of Grassland Livestock Industry Innovation, Ministry of Agriculture and Rural Affairs, Lanzhou, Gansu, China, **3** National Demonstration Center for Experimental Grassland Science Education, Lanzhou University, Lanzhou, Gansu, China, **4** College of Pastoral Agricultural Science and Technology, Lanzhou University, Lanzhou, Gansu, China

☯ These authors contributed equally to this work.
* yuanml@lzu.edu.cn

**Data Availability Statement:** The mitogenome sequences of two Formica species newly sequenced in this study have been deposited in NCBI (GenBank accession numbers ON408245,

## Abstract

*Formica* is a large genus in the family Formicidae with high diversity in its distribution, morphology, and physiology. To better understand evolutionary characteristics of *Formica*, the complete mitochondrial genomes (mitogenomes) of two *Formica* species were determined and a comparative mitogenomic analysis for this genus was performed. The two newly sequenced *Formica* mitogenomes each included 37 typical mitochondrial genes and a large non-coding region (putative control region), as observed in other *Formica* mitogenomes. Base composition, gene order, codon usage, and tRNA secondary structure were well conserved among *Formica* species, whereas diversity in sequence size and structural characteristics was observed in control regions. We also observed several conserved motifs in the intergenic spacer regions. These conserved genomic features may be related to mitochondrial function and their highly conserved physiological constraints, while the diversity of the control regions may be associated with adaptive evolution among heterogenous habitats. A negative AT-skew value on the majority chain was presented in each of *Formica* mitogenomes, indicating a reversal of strand asymmetry in base composition. Strong codon usage bias was observed in *Formica* mitogenomes, which was predominantly determined by nucleotide composition. All 13 mitochondrial protein-coding genes of *Formica* species exhibited molecular signatures of purifying selection, as indicated by the ratio of non-synonymous substitutions to synonymous substitutions being less than 1 for each protein-coding gene. Phylogenetic analyses based on mitogenomic data obtained fairly consistent phylogenetic relationships, except for two *Formica* species that had unstable phylogenetic positions, indicating mitogenomic data are useful for constructing phylogenies of ants. Beyond characterizing two additional *Formica* mitogenomes, this study also provided some key evolutionary insights into *Formica*.

https://www.ncbi.nlm.nih.gov/nuccore/ON408245;
ON408246, https://www.ncbi.nlm.nih.gov/nuccore/
ON408246).

**Funding:** This study was funded by the Second
Tibetan Plateau Scientific Expedition and Research
(STEP) Program (2019QZKK0302), and the Key
Project of Science and Technology Department of
Xinjiang Autonomous Region, China [2020E0213,
2016A03006]. The funders had no role in study
design, data collection and analysis, decision to
publish, or preparation of the manuscript.

**Competing interests:** The authors have declared
that no competing interests exist.

## 1. Introduction

Ants (Hymenoptera: Formicidae) are highly ecologically dominant organisms that typically
nest underground and play key roles in symbiotic interactions, soil aeration, and nutrient
cycling [1, 2]. There are currently over 14,106 extant ant species described worldwide, belong-
ing to 346 genera in 16 subfamilies (AntWeb, 2023). *Formica*, as a large genus in Formicidae,
is widely distributed and likely originated in Eurasia [3] Presently, 179 extant *Formica* species
are known, which are mainly distributed in Europe, Asia, most of North America, the Canary
Islands, and Morocco (AntWeb, 2023). Many species of *Formica* are widely used as biological
control agents, as they are characterized by their rapid reproduction and ease of introduction
and release [4, 5]. Ant phylogenetic relationships have been widely studied at various taxo-
nomic levels, and these previous studies consistently supported the monophyly of the Formici-
nae; however, the species relationships within Formicinae have remained controversial in
previous studies [6–8].

To adapt to more extreme habitats, such as low temperature, high altitude, and low oxygen
content, ants can evolve corresponding physiological mechanisms [9, 10]. Extreme habitat
conditions promote biodiversity, as beneficial alleles will be fixed by strong positive selection
and overwhelmed signatures of historical purifying selection [11, 12]. In addition to adaptive
physiological traits, there are also highly conserved ones, but the balance between the two is
not yet known [13]. In insects, highly conserved coding regions of mitochondrial genomes
(mitogenomes) may be important for ATP production, and their adaptation to different habi-
tats may mainly be reflected in the diversity of control regions (CRs) [14]. Insect mitogenomes
are approximately 16 kb in length and encode 37 genes: 2 rRNAs (*rrnL* and *rrnS*), 13 protein-
coding genes (PCGs), 22 transfer RNAs (tRNAs) [15]. The sequencing of complete mitogen-
omes is very important in the study of mitogenome architecture, evolutionary processes, phy-
logenetics, species identification, and management of invasive species [7, 16, 17]. The
comparative analysis of mitogenomes is a method commonly used to clarify the evolutionary
relationships among animals [18]. In recent years, advances in sequencing technology have
promoted the further development of ant mitogenomes. Up to now, mitogenomes have been
sequenced for 59 Formicidae species (GenBank, as of Mar 2023), but the sequenced *Formica*
mitogenomes in particular are still limited. This has also limited our understanding of mitoge-
nomic features and evolution within *Formica*.

To further explore the phylogeny and evolutionary characteristics of *Formica*, we sequenced
and annotated complete mitogenomes of *Formica candida* and *F. glauca*, and performed a
comparative mitogenomic analysis for 12 *Formica* species, focusing on general mitogenomic
features, codon usage, evolutionary characteristics of PCGs, base composition, tRNA struc-
tures, and conserved elements within both large CRs and small intergenic regions. Through
comparative analysis, we found many conserved mitochondrial features within *Formica*. We
also reconstructed species-level *Formica* phylogeny based on mitogenomic data using three
analytical methods (maximum likelihood [ML], neighbor-joining [NJ], and Bayesian inference
[BI]). By linking the relationship between the mitogenomic characteristics of *Formica* with the
results of previous studies and phylogenetic analysis, the present study shows the effectiveness
of mitogenomic approaches to phylogenetics.

## 2. Materials and methods

### 2.1. Sampling and DNA extraction

Adult specimens of *F. candida* and *F. glauca* were collected from Qumalai County, Qinghai
Province, China and Altay City, Xinjiang Uygur Autonomous Region, China, respectively (S1

Table). The two *Formica* species used in this study are not included in the "List of Protected Animals in China" and no ethical permissions were required for field samping. All samples were initially preserved in 100% ethanol at the sampling site and then stored at -80˚C. Total genomic DNA was extracted from a single specimen of each species using a DNeasy Tissue Kit (Qiagen, Germany). We evaluated the quality of the extracted genomic DNA by using 1.5% agarose gel electrophoresis and the NanoDrop spectrophotometer (Thermo Scientific, Waltham, MA, USA).

## 2.2. Mitogenome sequencing, assembly, and annotation

The entire mitogenome sequences of the two *Formica* species were sequenced by using the Illumina NovaSeq 6,000 platform (Illumina, San Diego, CA, USA) with 150-bp paired-end reads, conducted by Wuhan Benagen Tech Solutions Co., Ltd. (Wuhan, China). We removed low-quality reads by using SOAPnuke 2.1.0 [19], and the remaining reads (high-quality reads) were used to assemble the mitogenomes by using SPAdes 3.13.0 [20]. The two assembled mitogenomes were annotated by using MITOS (http://mitos2.bioinf.uni-leipzig.de) [21] to identify each of the 37 mitochondiral genes by using the mitogenomes of *Formica* available in GenBank as references. Tandem repeats within the CRs were detected by using the Tandem Repeats Finder web (https://tandem.bu.edu/trf/trf.html). We used Mfold (http://www.mfold.org/) to construct potential secondary structures of larger gene intervals. The two *Formica* mitogenomes newly sequenced in this study have been deposited in NCBI (GenBank accession numbers ON408245-46).

## 2.3. Comparative mitogenomic analysis

We used MEGA X [22] to analyze the mitochondrial nucleotide composition and codon usage of 12 *Formica* species. Strand asymmetry was evaluated by calculating AT-skew and GC-skew values with the method: AT skew = $[A − T]/[A + T]$ and GC skew = $[G − C]/[G + C]$ [23]. We calculated the codon bias index (CBI) and the effective number of codons (ENC) for the 13 PCGs of each *Formica* mitogenome by using DnaSP [24]. We also calculated nucleotide composition of the first, second, and third codon positions of 13 PCGs using CUSP (https://www.bioinformatics.nl/emboss-explorer/). To further investigate the codon usage bias among the 12 *Formica* species, we analyzed relationships between ENC, CBI, G + C content of all codon positions, and G + C content of the third codon positions. We used the ENC curve to determine the dominant evolutionary force for shaping the codon usage bias of the mitochondrial PCGs. The actual ENC values are all below an ENC curve, indicating that the dominant factor of variation is natural selection; otherwise, mutation is the dominant factor [25]. The values of nonsynonymous substitutions per nonsynonymous site ($K_a$) and synonymous substitutions per synonymous site ($K_s$) for each PCG were calculated by using MEGA X [26].

## 2.4. Phylogenetic analysis

Phylogenetic analyses were performed using mitogenomic data from 12 *Formica* species and species from two other Formicinae genera (S2 Table). The species *Myrmica scabrinodis* (NC_026133) from Myrmicinae was used as the outgroup. The sequences of PCGs were aligned using Clustal W (Codon) in MEGA X [22], and PCGs were translated employing the standard invertebrate mitochondrial genetic code. Two mitogenomic datasets were used for phylogenetic analyses: i.e. the P123 dataset (nucleotide sequences, all codon sites of 13 PCGs, including 11,184 nucleotides in total) and the P123AA dataset (inferred amino acid sequences of 13 PCGs, including 3,728 amino acids in total). Potential sequence saturation in our alignments were evaluated by using a substitution saturation test, impletemented in DAMBE 5.3.74

[26], and there was no substantial substitution saturation (S3 Table). The best partitioning schemes and corresponding evolutionary models for the two datasets were selected by IQ-TREE (S4 Table) and used for the following phylogenetic analyses.

We performed ML phylogenetic analyses by using RaxML-HPC2 [27], with the GTR+Γ model and 1,000 bootstraps (BS). BI analyses were conducted with MrBayes 3.2.7 [28], running $1 \times 10^8$ generations with sampling every 100 generations. NJ trees were constructed by using MEGA X [22], with the Kimura two-parameter molecular evolutionary model.

## 3. Results

### 3.1. General features of *Formica* mitogenomes

We obtained the complete mitogenomes of *F. candida* and *F. glauca* (S1 Table). The two newly sequenced mitogenomes encoded all the 37 typical mitochondrial genes and contained a CR. Twenty-three genes (9 PCGs and 14 tRNAs) were encoded on the majority strand (J-strand), whereas the remaining 14 genes on the minority strand (N-strand). The gene arrangement was conserved within all sequenced *Formica* species, but differed from that of the ancestral insect mitogenome, with *trnM* showing a translocated position in each of the seven completely sequenced *Formica* species.

The complete mitogenomes of seven *Formica* species displayed difference in size, ranging from 16,492 bp in *F. glauca* to 17,432 bp in *F. sinae* (Fig 1). This difference was primarily owing to size variation of the CRs, ranging from 399 bp in *F. glauca* to 1331 bp in *F. neogagates* (Fig 1). Of these seven species, the largest intergenic regions were mainly located between *trnF* and *nad5*, as the largest gene overlap regions were primarily between *atp8* and *atp6*. All the 22 tRNAs presented a typicall cloverleaf structure (i.e. four arms), except for two tRNAs (*trnS1* and *trnE*). *trnS1* lost the dihydrouridine (DHU) arm, whereas *trnE* lacked the TΨC stem in both *F. candida* and *F. glauca* (Fig 2).

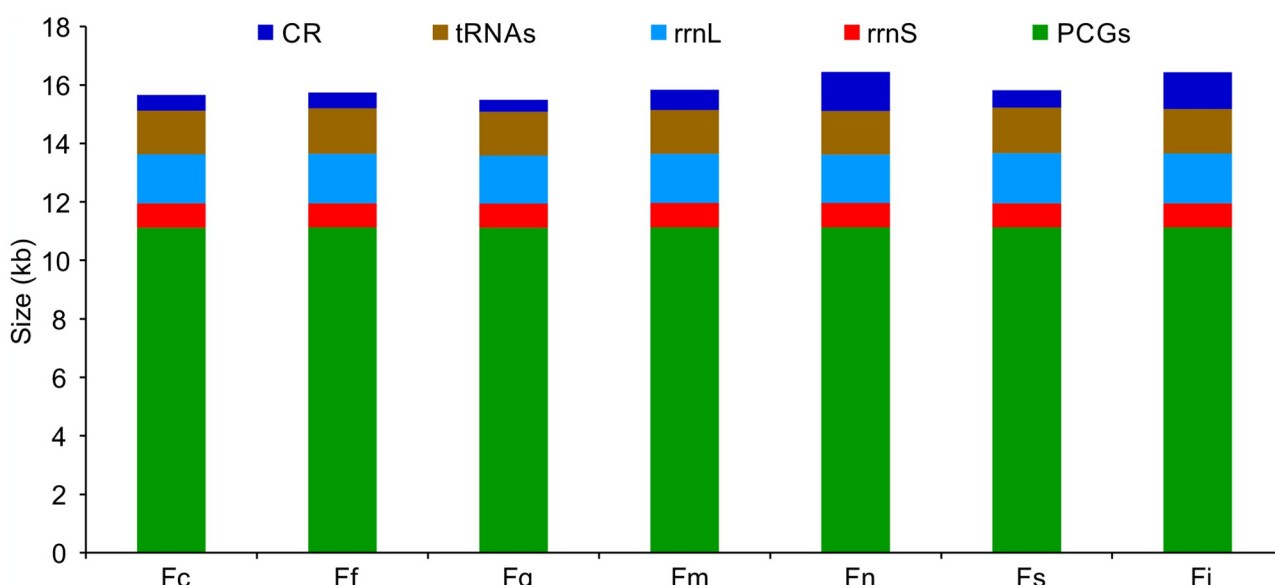

**Fig 1. The size of protein-coding genes (PCGs), tRNA, *rrnL*, *rrnS*, and control region (CR) sequences among *Formica* mitochondrial genomes.**
Ant species names are abbreviated as follows: *Formica candida*, *Fc*; *Formica fusca*, *Ff*; *Formica glauca*, *Fg*; *Formica moki*, *Fm*; *Formica neogagates*, *Fn*; *Formica selysi*, *Fs*; *Formica sinae*, *Fi*.

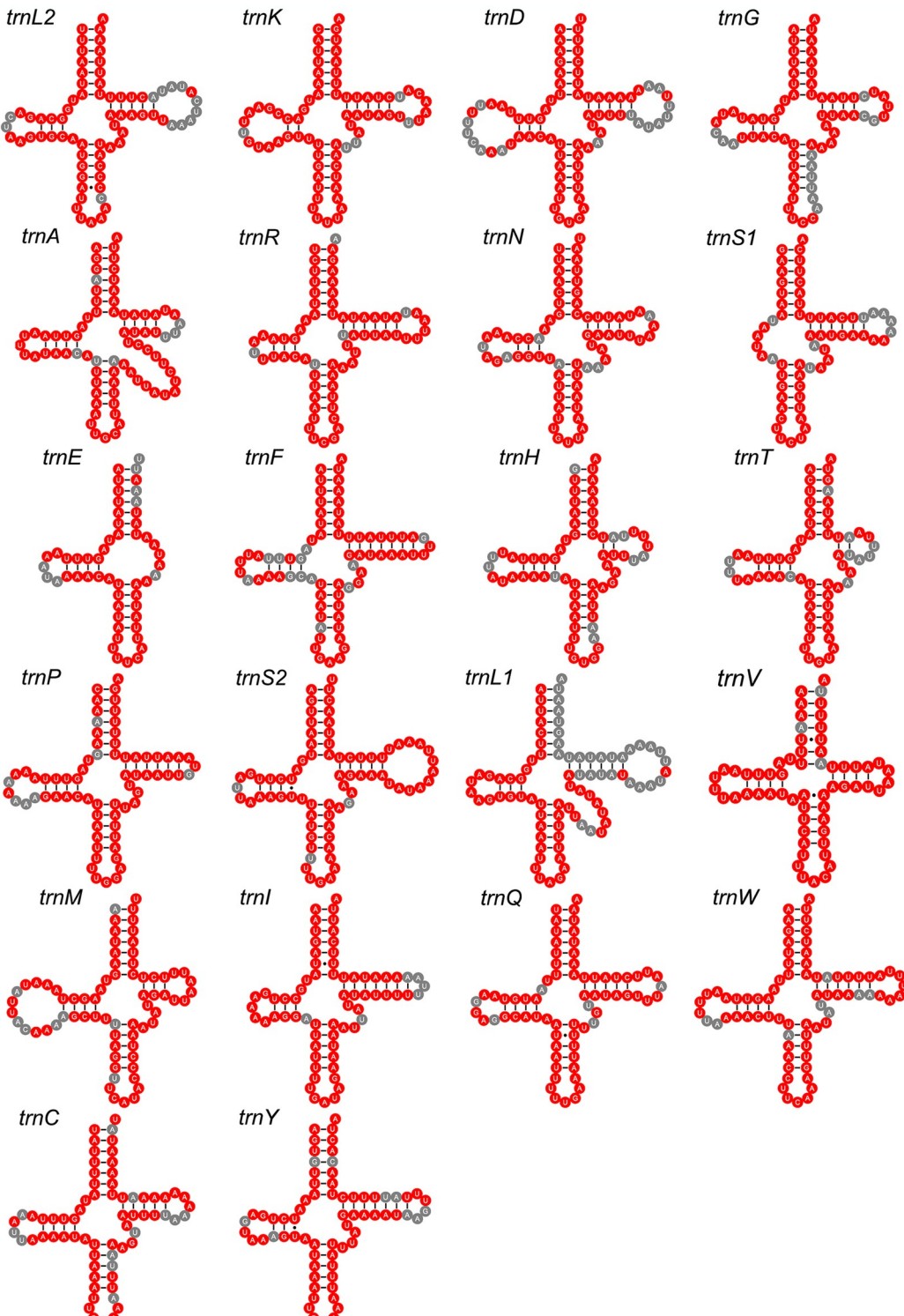

**Fig 2. Putative secondary structures of the 22 tRNA genes found in the *Formica candida* mitogenome.** All tRNA genes are shown in the order of occurrence in the mitochondrial genome starting from *trnL2*. Completely conserved sites within the twelve species are shown as white nucleotide abbreviations within red spheres. Bars indicate Watson–Crick base pairings or G and U pairs. Unpaired bases are represented as dots.

## 3.2. Nucleotide composition and codon usage

The base composition of the two newly sequenced *Formica* mitogenomes was enriched in A and T, with the A+T content of 83.8% in *F. candida* and 83.4% in *F. glauca*. High A+T content (>81%) was also observed in the other ant mitogenomes (Fig 3A). The *Formica* mitogenomes exhibited a negative GC-skew value, with a moderate average value (-0.312 ± 0.01) (Fig 3B), while all sequenced *Formica* mitogenomes exhibited a slightly negative AT-skew, ranging from -0.033 to -0.003 (Fig 3A). Codon numbers of *Formica* mitogenomes ranged from 3,274 in *Formica* sp.DM656 to 3,712 in *F. rufa* (S5 Table). Relative synonymous codon usage (RSCU) analysis revealed that the two *Formica* species (*F. sinae* and *F. moki*) used all the 62 mitochondrial codons, while the remaining 10 *Formica* species did not use one or two codon (S5 Table). Fifty-six codons were consistently used in all the 12 *Formica* mitogenomes, and four AT-rich codons (UUU [F], AUU [I], AUA [M], and UUA [L]) were the most commonly used (S5 Table). However, the frequency of GC-rich codons was low, especially for CGC (R), GCG (A), and CGG (R), which were not used in at least one species.

The average ENC value among all the PCGs was 34.19, ranging from 33.04 (in *F. candida* and *F. podzolica*) to 35.17 (in *F. sinae*). We found positive correlations between G + C content for all codon positions and ENC ($R^2 = 0.79$, $P < 0.01$; Fig 4A), so was between G + C content for the third codon positions and ENC ($R^2 = 0.99$, $P < 0.01$; Fig 4B). Negative correlations were found between CBI and both G + C content for all codon positions ($R^2 = 0.96$, $P < 0.01$; Fig 4C) and G + C content of the third codon positions ($R^2 = 0.97$, $P < 0.01$; Fig 4D), so was CBI and ENC ($R^2 = 0.98$, $P < 0.01$; Fig 4E). The observed ENC values for all *Formica* species were below the ENC curve (Fig 5A), and no significant correlation ($R^2 = 0.03$, $P > 0.05$) was found between the combined GC content of the first and second codon positions and the GC content of the third codon positions (Fig 5B), indicating that codon usage bias in *Formica* mitogenomes might be influenced by natural selection.

## 3.3. Intergenic spacers

The *Formica* mitogenomes contained intergenic spacers (IGSs) of varying lengths, abundantly dispersed through almost all of the genes and with extremely high A+T contents. The A+T contents of the two newly sequenced mitogenomes were 93.78% (*F. glauca*) and 94.09% (*F. candida*). Here, we mainly describe IGSs with conserved sequences or notable structures. Although individual IGSs differed in length among species, their sequences were highly conserved. All of them showed two key characteristics, i.e., AT-enrichment and conserved sequences. In addition, some IGSs also contained microsatellites. Regarding the secondary structure, it was found that some IGSs had stem-loop structures, i.e., the IGSs between *trnQ* and *nad2*, *cox2* and *trnK*, *atp6* and *cox3*, *cox3* and *trnG*, *trnS1* and *trnE*, *trnF* and *nad5*, *nad4L* and *trnT*, and also *cob* and *trnS2* (S1 Fig).

Although the sequences of some IGSs were very short (<20 bp), they were highly conserved, i.e., *trnI–trnQ* (TAADTWA) (S2 Fig), *trnH–nad4* (WTAWAAA) (S2 Fig), *trnS2–nad1* (TAAATTAYA) (Fig 6). In addition, a total of 14 relatively long IGSs were present in the *Formica* mitogenomes (S2A–S2R Fig). Highly conserved regions of the 14 IGSs were found among all *Formica* species, some of which included microsatellites. We also found that several IGSs (e.g. *cox1-trnL2*, *cox2-trnK*, *atp6-cox3*, *cox3-trnG*, *trnN-trnS1*, *trnF-nad5*; S2 Fig) were more conserved in sequence size and similarity among five *Formica* species (i.e. *glauca*, *sinae*, sp.DM659, sp.DM658, and sp. DM656) which clustered together in phylogenetic tree (see the following section 3.6).

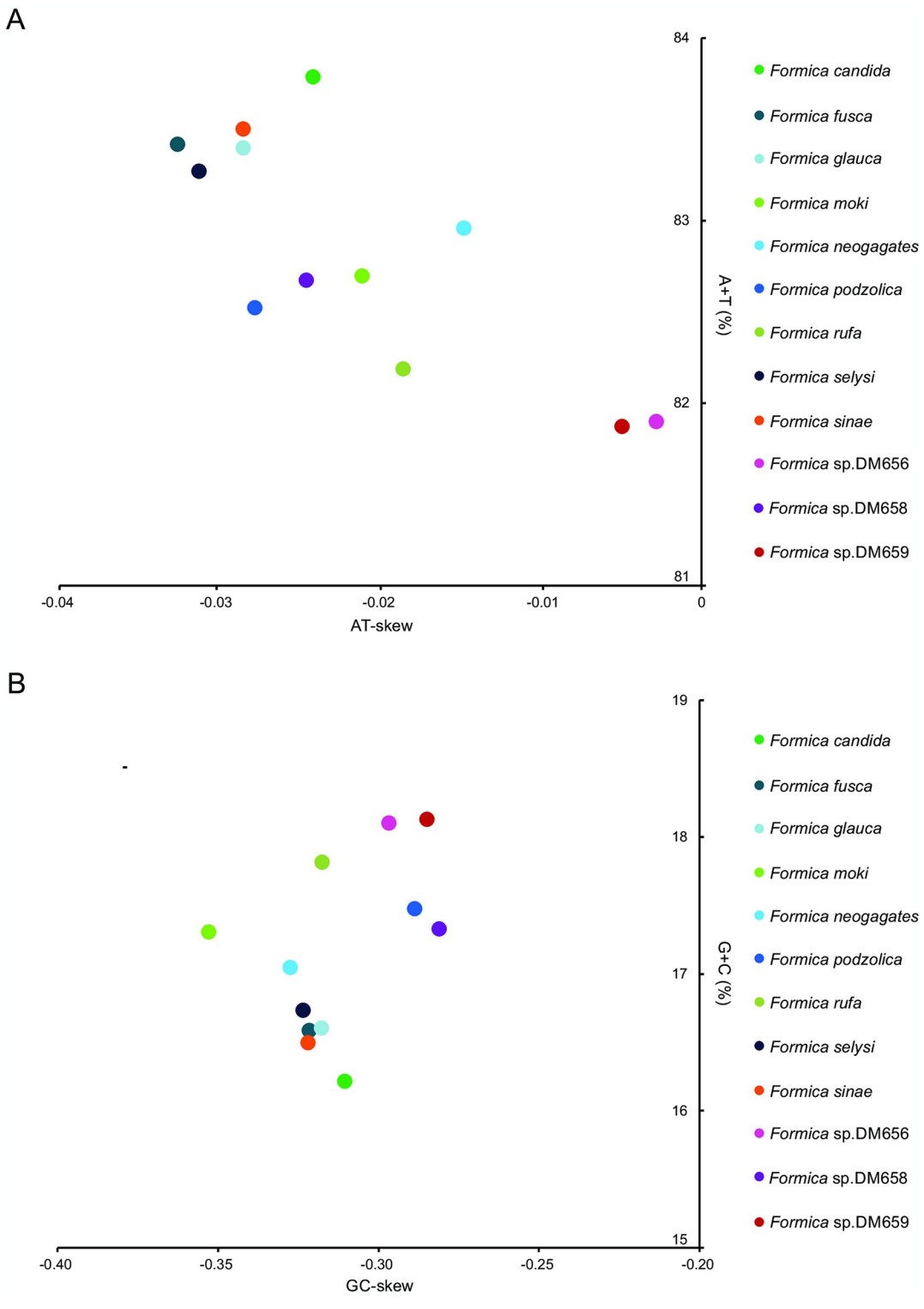

**Fig 3. AT% versus AT-skew and GC% versus GC-skew in the 12 *Formica* mitochondrial genomes.** Measured in bp percentage (*y*-axis) and level of nucleotide skew (*x*-axis). Values are calculated for J-strands in full-length mitochondrial genomes. A, A + T% vs AT-skew; B, G + C% vs GC-skew.

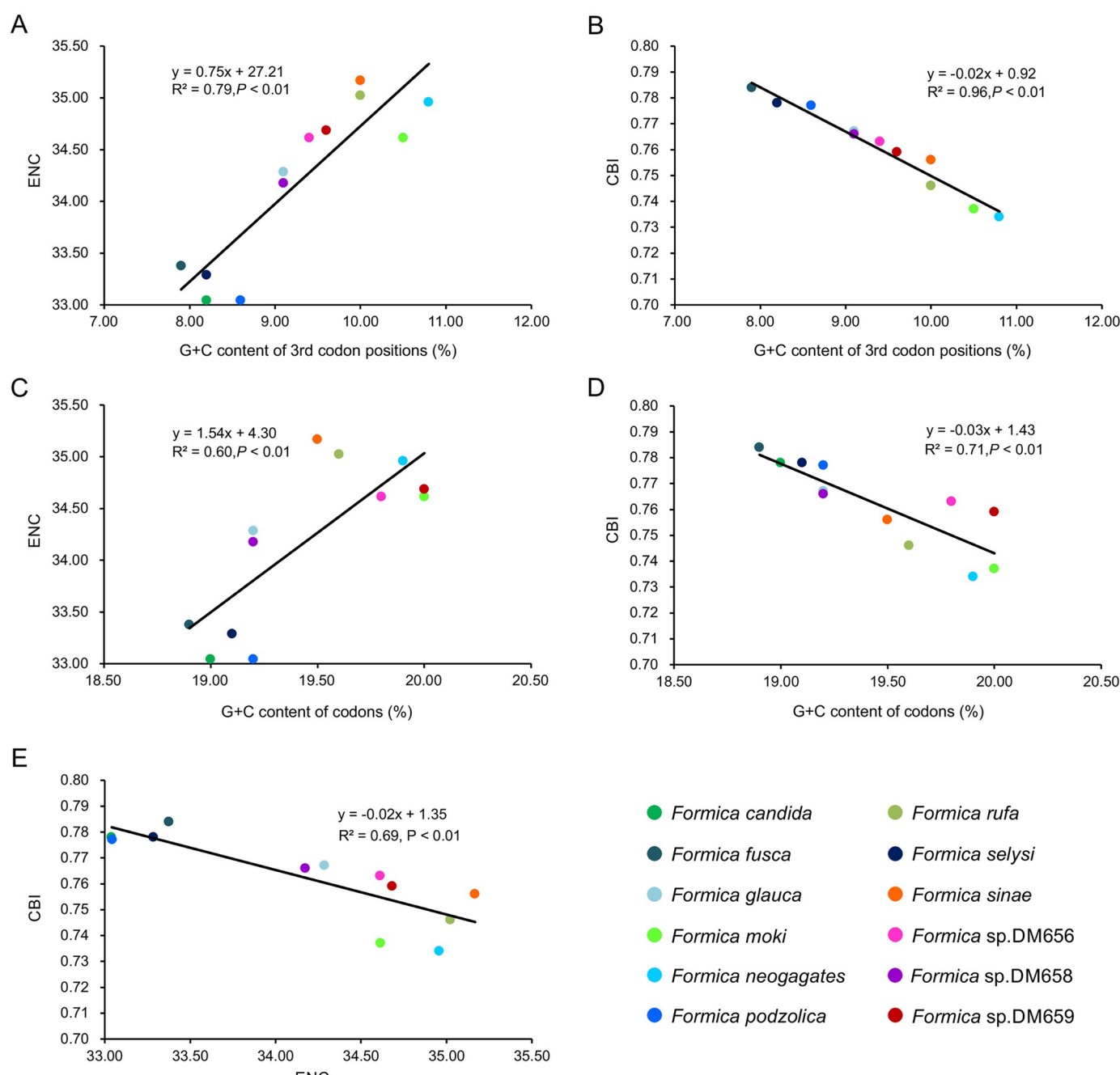

**Fig 4. Evaluation of codon bias in the mitochondrial genomes of 12 *Formica* species.** G + C%, G + C content of all codon positions; (G + C)3%, G + C content of the third codon positions; ENC, effective number of codons; CBI, codon bias index.

## 3.4. Control region

Eight *Formica* mitogenomes with CR sequences contained only one CR, which was located between *rrnS* and *trnM*. The AT contents of the CRs of eight mitogenomes ranged from 82.41% (*F. moki*) to 92.36% (*F. candida*). Among these eight mitogenomes of *Formica* species, all CRs had typically high A+T contents. We observed some essential components among the

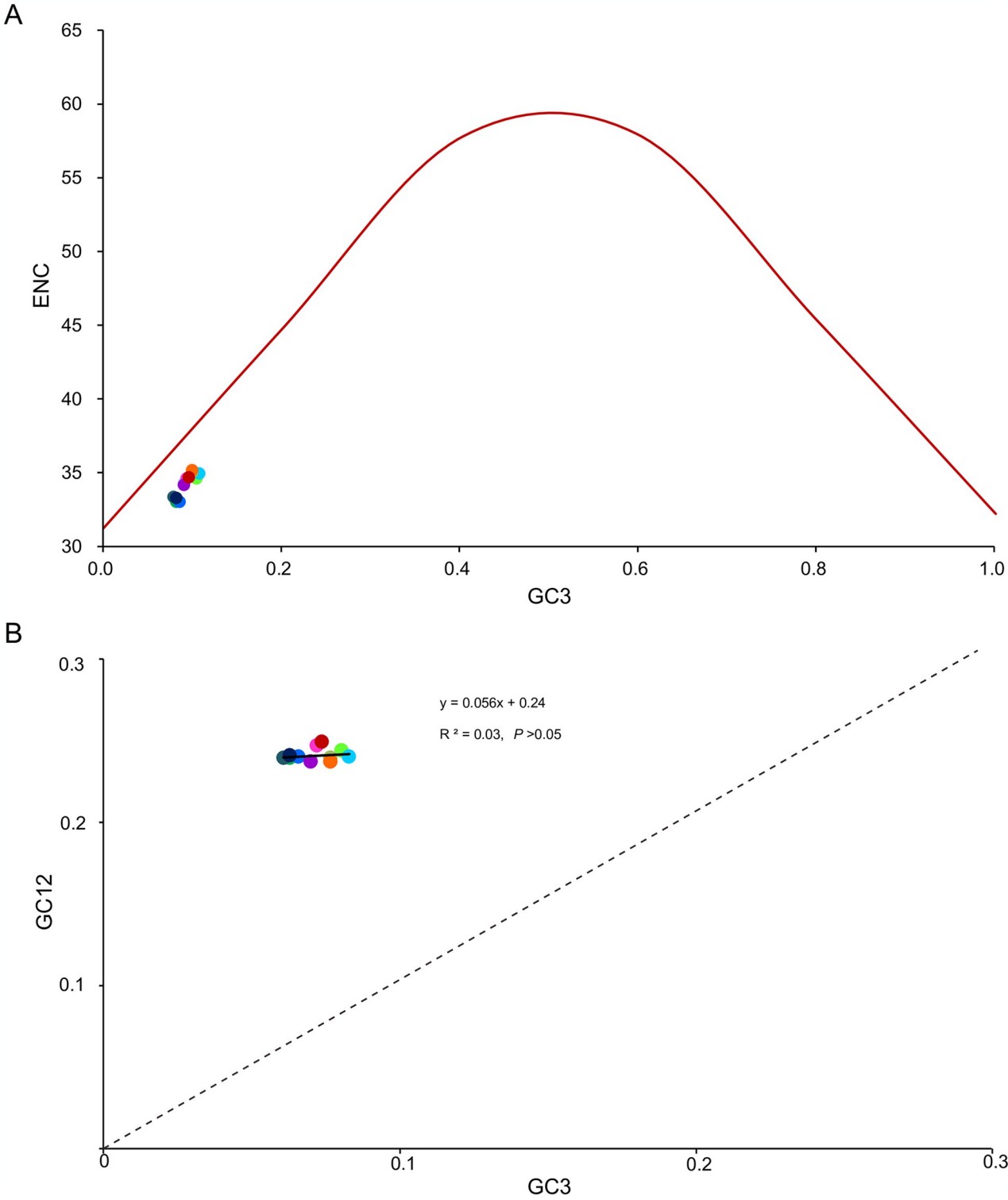

**Fig 5. The correlation between effective number of codons (ENC) and G + C content of the third codon positions (GC3) for 12 *Formica* species.**
The colored dots correspond to those in Fig 3. (A) The solid line represents the relationship between ENC and GC3 content. (B) The solid line represents the relationship between GC12 and GC3 content, whereas the dotted line indicates *y = x*. GC12, G + C content of the first and second positions.

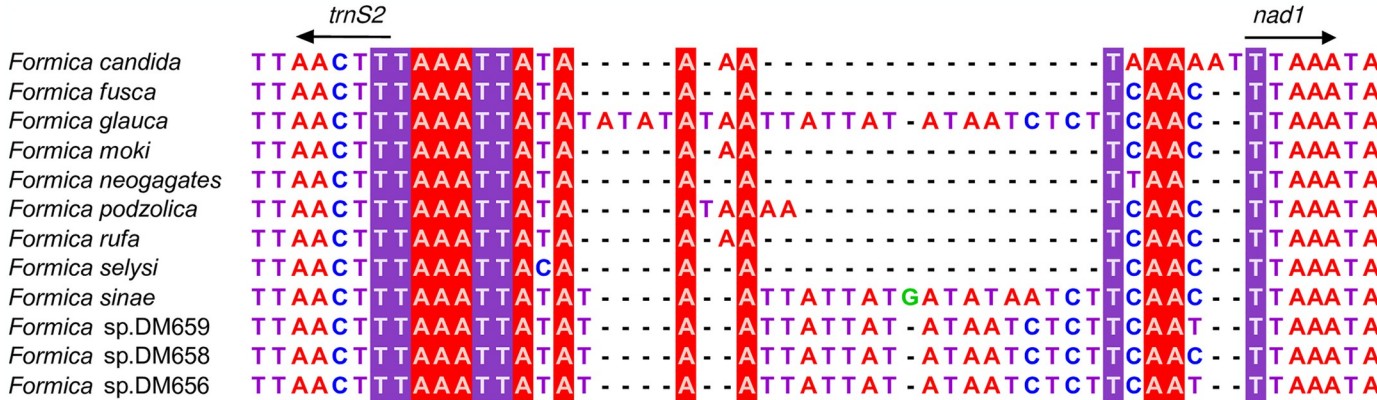

**Fig 6. Sequence alignment of the intergenic spacer between *trnS2* and *nad1* for *Formica* mitochondrial genomes.**

*Formica* mitogenomes: (1) AT-rich regions; (2) poly-T sequences (except in *F. moki*); (3) poly-A sequences (in *F. candida*, *F. fusca*, and *F. selysi*). In addition, we found a large number of TATA motifs as well as large tandem repeat units in CRs. Five tandem repeat units presented in the CR of *F. neogagates*, whereas the CRs of six *Formica* species had two or three tandem repeat units.

### 3.5. Protein-coding genes

All the 13 PCGs began with typical codons (ATN) and terminated with a complete stop codon (TAA) in the mitogenomes of *F. candida* and *F. glauca*, whereas other *Formica* species did often have incomplete T or TA stop codons. Among all of the 13 PCGs, A + T content (89.2–92.1%) of the third codon position was higher than that of the first (76.5–77.7%) and second (73.1–74.0%) codon positions (Fig 7). The average values of both $K_s$ and $K_a$ in the 12 *Formica*

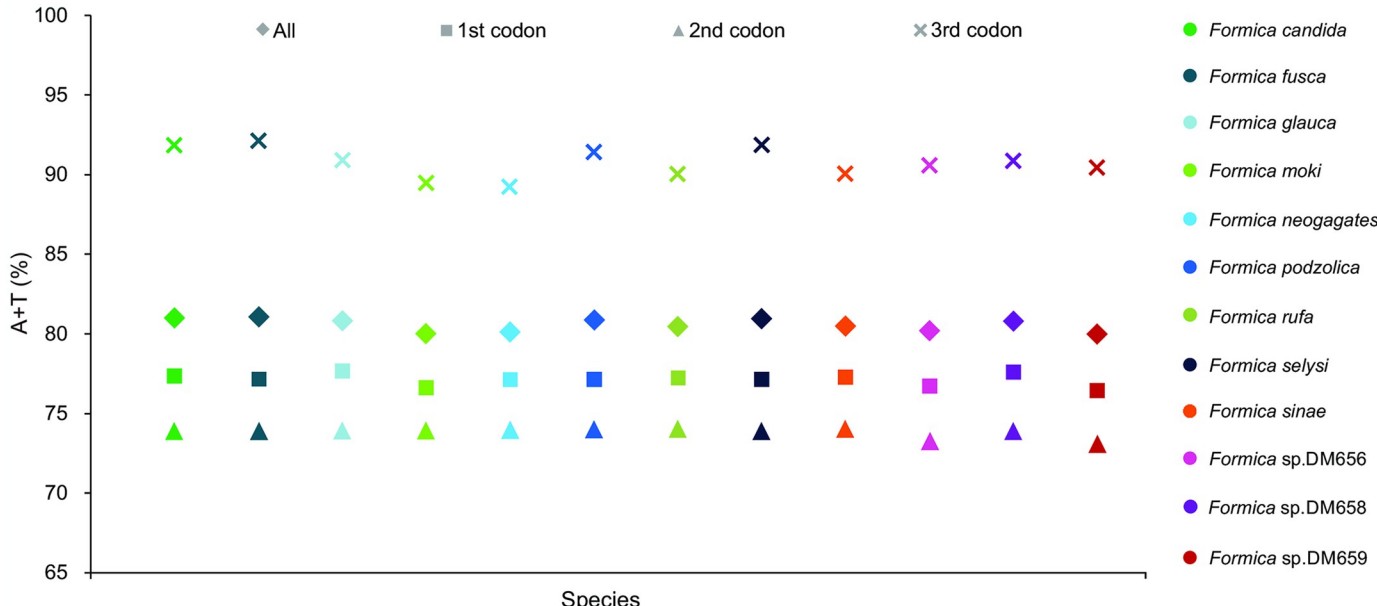

**Fig 7. A + T contents of the mitochondrial protein-coding genes in *Formica* mitochondrial genomes.**

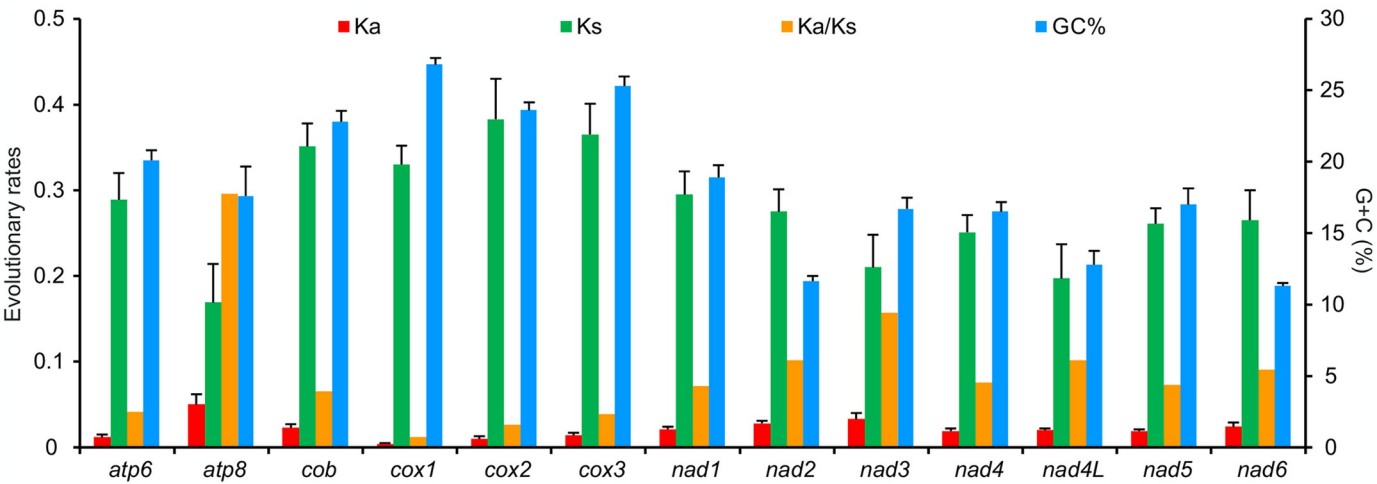

**Fig 8. Evolutionary rates of 13 protein-coding genes in the mitochondrial genomes of 12 species of *Formica*.** The left *y*-axis shows the substitution rate of mitochondrial genes, while the right *y*-axis shows the G + C content. Synonymous nucleotide substitutions per synonymous site ($K_s$) and nonsynonymous nucleotide substitutions per nonsynonymous site ($K_a$) were calculated using the Kumar method. The standard error estimates were obtained by a bootstrap procedure (1,000 replicates).

species differed among 13 PCGs (Fig 8), indicating that the mutation rate was relatively low. The $K_a/K_s$ values also varied considerably among the 13 PCGs of the 12 *Formica* species and were less than 1 (Fig 8). The $K_a/K_s$ values of *atp8* (0.296) was the largest, indicating a fastest evolutionary rate of *atp8*. Two PCGs (*nad2* and *nad3*) also showed more amino acid substitutions, whereas *cox1* was the most conserved (Fig 8).

### 3.6. Mitochondrial phylogeny of *Formica*

Two datasets (P123 and P123AA) and three methods (ML, BI, and NJ) resulted in six phylogenetic trees with highly similar topologies (Fig 9 and S3 Fig). The two phylogenetic topologies differed only in the phylogenetic position of *F. candida* and *F. rufa*. All the six phylogenetic trees consistently supported the monophyly of *Formica* and *Cataglyphis*, with high support values (Fig 9 and S3 Fig).

All the phylogenetic trees supported of the following phylogenetic relationship: (((*Formica* + *Polyergus*) + *Cataglyphis*) + *Myrmica scabrinodis*). Among *Formica* species, *F. neogagates* was sister to the remaing 11 species which were divided into two phylogenetic groups: group 1 including six species, and group 2 including five species. Within group 1, *F. fusca* first clustered with *F. selysi*, which both further clustered with *F. podzolica*, with *F. moki* at the base of group 1. The phylogenetic positions of *F. candida* and *F. rufa* were unstable. Within group 2, *F. glauca* was located between *F. sinae* and the three other *Formica* species in the group.

## 4. Discussion

### 4.1. General features of *Formica* mitogenomes

The two newly sequenced *Formica* mitogenomes had typical gene contents that were identical to those of other sequenced ant [29, 30] and insect mitogenomes [15]. The observed gene rearrangement that occurred in *Formica* has been previously reported in the family Formicidae [25]. The rearrangement of Formicidae is consistent with a duplication/random loss model [31, 32]. Therefore, this rearrangement of the *Formica* mitogenomes can be explained by the

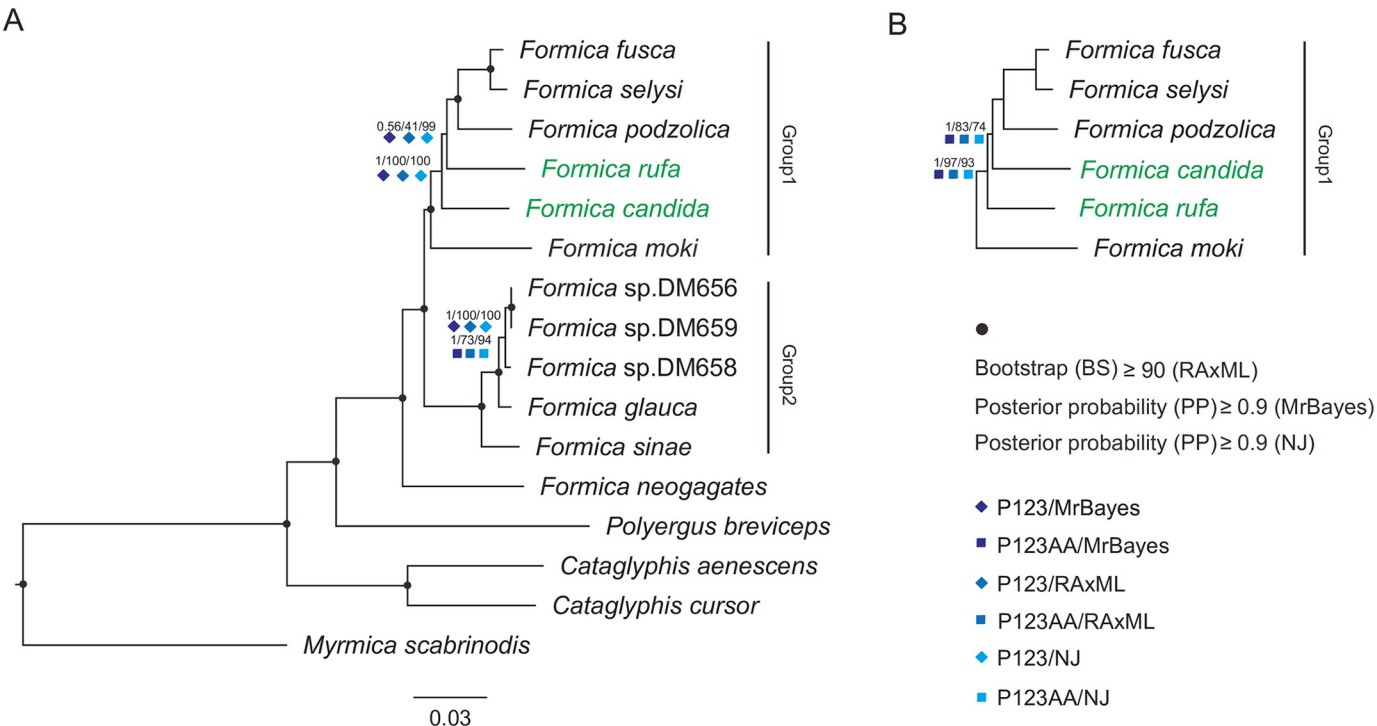

**Fig 9. Two phylogenies of 16 Formicinae species from four genera based on two datasets (P123 dataset and P123AA dataset) and three analytical methods (Bayesian inference [BI], neighbor-joining [NJ], and maximum likelihood [ML]).**

plesiomorphic *trnI-trnQ-trnM* sequence and tandem duplication, as it may be owing to the tandem duplication of *trnI-trnQ-trnM* and subsequent loss of the first *trnI-trnQ* and the second *trnM*, eventually resulting in the observed *trnM-trnI-trnQ* sequence. Unlike the anticodons of *trnS1* and *trnK* in most insects, the *Formica* genus specifically uses TCT and TTT, and it has been found that those abnormal anticodons are related to gene rearrangement [33]. One common feature of insect mitogenomes is that *trnS1* lacks a DHU arm [34, 35]. However, *trnE* lacking the TΨC stem is not common in insects, but is found in other arthropods, such as spiders [36, 37]. In addition, we analyzed the conservation of the secondary structure of tRNAs in 12 *Formica* species. We found these structural nucleotides of the tRNAs were comparatively conserved, and the stem was more conserved than the ring of corresponding tRNAs apart from the difference in the anticodon loop. These conserved regions may be association with the structure and function of tRNAs [38].

## 4.2. Nucleotide composition and codon usage

Insect mitogenomes generally show a positive value for AT-skew and a negative value for GC-skew on the J-strand. However, our sequenced *Formica* mitogenomes presented a slightly negative AT-skew, which indicated that the incidence of Ts was higher than that of As, and similar results have been reported in insects such as Galleriinae [39], *Apostictopterus fuliginosus* [40], and leaf hopper [41]. This negative AT-skew may be associated with codon positions, gene direction, and replication, e.g., probabilities of nucleotide repair differing during the replication process following damage to single-standard DNA [42, 43].

Codon usage bias is an important evolutionary phenomenon commonly found in many animals. Codon usage bias is mainly driven by the frequency of synonymous codons used in the coding region of the mitogenomes differing. Other many factors also could affect codon usage bias, e.g., selection for optimized translation, gene expression, codon location within genes, and the secondary structural of DNA [44]. Generally, mutation pressure and natural selection are considered to be two main factors affecting codon usage [45, 46]. As was found in other insects [46, 47], our results of the RSCU analyses also indicated that the third codon positions had a higher frequency in the usage of A and T relative to G and C, which may have led to the high codon bias observed. The negative correlation between CBI and ENC indicated that the reduced ENC could lead to high codon usage bias [48]. We proposed that differences in codon usage bias of *Formica* mitogenomes might be influenced by both natural selection and mutation pressure, as has also been reported in other insects [46, 48].

## 4.3. Evolutionary rates of protein-coding genes

Estimating the $K_a/K_s$ value of PCGs has been widely used to indicate how natural selection affect sequence evolution in various animals [49], i.e. $K_a = K_s$ indicating neutral mutation, $K_a/K_s < 1$ indicating purifying selection, and $K_a/K_s > 1$ indicating positive selection [50]. The values of $K_a/K_s$ for all the 13 PCGs in *Formica* mitogenomes were less than 1, indicating that these mitochondrial genes might be evolving under purifying selection [51, 52]. As purifying selection eliminates harmful that arise mutations, it may thus dominate the evolution of mitogenomes [53]. There was a negative correlation between the $K_a/K_s$ values of the 13 PCGs and the G + C content ($R^2 = 0.73$, $P < 0.01$), indicating that differences in G + C content may lead to different evolutionary patterns of PCGs in *Formica* mitogenomes. Strong purifying selection and low mutation rates dominate mitochondrial genome evolution [54]. The $K_a/K_s$ value of *cox1* presented a slowest evolutionary rate, as has been reported in many insect mitogenomes [55], suggesting fewer changes in amino acids and the conservation of this gene [56].

Most ants build nests underground, which can provide protection from predators and extreme weather, but reduces oxygen concentrations and causes and have high levels of carbon dioxide accumulation [10, 57]. This hypoxic environment may impose relatively stronger purifying selection pressure of subterranean lineages [10]. Most genes have greater $K_a/K_s$ values in subterranean lineages than in non-subterranean lineages [58]. However, the $K_a/K_s$ values of *Formica* were lower than that of other hymenopterans [59, 60]. This may be a highly conserved physiological defense character that has evolved in *Formica* ants as an adaptation to this hypoxic environment, thus ensuring the normal functioning of mitogenome [13].

## 4.4. Non-coding region

The lack of conservation of repeat units among these *Formica* mitogenomes may be associated with the size variation of CRs and functional lack of these repeat units [61]. The CR of insect mitogenomes plays a key role in both transcription initiation and replication process of mitochondrial genes [62, 63]. The position of the CR between *rrnS* and *trnM* in *Formica* mitogenomes was consistent with that of other ants [17, 63], indicating conservation in the number and location of the *Formica* CRs. In eight *Formica* species, all CRs had typically high A+T contents, and the types of base substitutions that can occur are limited compared to the those that can occur in other regions [64, 65]. We observed some essential components among the *Formica* mitogenomes, as has been reported [66]. However, tandem repeat sequences and poly-T and poly-A regions were not found in the CRs of some *Formica*, and tandem repeat sequences differed in *Formica* species. These characteristics indicate the diversity in CR structures in the *Formica* mitogenomes, and the variation in CR length may be the result of variable numbers of

tandem repeats [67]. We also found many stem-loop structures in the CRs, and some stem-loop structures may be associated with the initiation of replication and transcription.

### 4.5. *Formica* phylogeny and intergenic spacers

The use of mitogenomic data is a common approach to exploring phylogenetic relationships among different insect groups [68, 69]. The sister relationship between *Formica* and *Polyergus* ants confirmed in the present work was largely congruent with the results of previous studies [6]. Among the two major groups inferred for *Formica* ants, the consistent phylogenetic relationship between the four species (*F. fusca*, *F. selysi*, *F. rufa* and *F. candida*) in group 1 has also been supported by other research [70]. However, another phylogenetic relationships of the five species within group 1 was inferred in previous studies, supporting a sister relationship of *F. rufa* and *F. candida* [8, 71]. Mitogenomic sequences have been extensively used for reconstructing phylogene in many animals [72–75]. In the present analyses, two different tree topologies were obtained, indicating that phylogenetic results can be potentially influenced by both the mitogenomic datasets and inference methods. This unstable phylogenetic relationships of *Formica* were also reported in previous studies. Although there were a few different relationships within *Formica* based on different datasets, all analyses supported the relationship of (((*Formica* + *Polyergus*) + *Cataglyphis*) among the three different ant genera. Considering the limited species sampled in this study, sequencing more *Formica* mitogenomes is needed to improve our understanding of *Formica*.

The IGSs of *Formica* mitogenomes varied in size, lacked repeat units, and were abundantly dispersed between genes, and changes in the size of IGSs are considered to be a shared derived trait of social insects [76]. However, the individual IGSs had one or more conserved regions among species, and the nucleotide composition of these IGSs was similar to that of adjacent genes. For example, regarding the nucleotide composition of the IGS between *nad6* and *cob*, the G + C content of this IGS was 18.3%, while the G + C content of *nad6* was 30.2%, suggesting that this sequence may have been derived from *nad6*. The evolutionary mechanism of IGSs may be explained by the slipped-strand mispairing and the duplication/random loss model [35, 77]. A conserved motif was located in the IGS of *trnS2-nad1*, which has been predicted to be the binding site of a mitochondrial transcription termination factor (DmTTF) [78]. The similar conserved motif has been widely reported in various insect mitogenomes [73, 79]. A 7-bp conserved motif (TAAATTA) presented in *Formica* mitogenomes was higly similar to the conserved motif (THACWW) in Hymenoptera [80].

In addition, we have linked the sequences and structures of the IGSs with phylogenetic relationships in *Formica*, demonstrating that this feature contributes to a phylogenetic understanding of the genus *Formica* [81, 82]. The following three examples are rather illustrative. The IGS between *atp6* and *cox3* genes was relatively conserved, except in *F. neogagates*. This indicated that this species had a distant genetic relationship with other *Formica* species, corresponding to inferred phylogenetic relationships. There was only one conserved sequence between *trnF* and *nad5* in the 12 species analyzed. When comparing *F. glauca*, *F. sinae*, *Formica* sp.DM659, *Formica* sp.DM658, and *Formica* sp.DM656, we found that the sequence of this IGS was almost completely conserved (S2O Fig), while this same IGS was also relatively conserved between *F. candida*, *F. fusca*, *F. moki*, *F. podzolica*, *F. rufa*, and *F. selysi* (S2P Fig). Compared to other *Formica* species, *F. fusca* and *F. selysi* had a larger IGS (>50 bp) between *trnM* and *trnI*, and the two species had the closest genetic relationship in the phylogenetic tree. Thus, we find the distinctive feature of the IGS regions and phylogenetic relationships to be valuable for a systematic understanding of the genus *Formica*.

## 5. Conclusion

We sequenced the complete mitogenomes of *F. candida* and *F. glauca*, further expanding the number of sequenced Formicidae mitogenomes. These two mitogenomes were similar in size to those of other ants. *Formica* mitogenomes were highly conserved in gene arrangement, gene content, nucleotide composition, codon usage, and PCG evolutionary patterns. Phylogenetic relationships within Formicinae obtained here were similar to previously inferred relationships, suggesting that mitogenomic data could be usefull for resolving the ant phylogeny. This study provides valuable insights into the phylogenetic relationships of *Formica*. Sequencing more mitogenomes across various taxonomic levels will greatly improve our understanding of both phylogenetic relationships and key subjects relevant to ants, such as the evolution of their strategies in behavior and life history.

## Supporting information

**S1 Fig. Stem-loop structures of intergenic spacers in *Formica* mitochondrial genomes.** (A) The intergenic spacer between *trnQ* and *nad2*. (B) The intergenic spacer between *cox2* and *trnK*. (C) The intergenic spacer between *atp6* and *cox3*. (D) The intergenic spacer between *cox3* and *trnG*. (E) The intergenic spacer between *trnS1* and *trnE*. (F) The intergenic spacer between *trnF* and *nad5*. (G) The intergenic spacer between *nad4L* and *trnT*. (H) The intergenic spacer between *cob* and *trnS2*.
(TIF)

**S2 Fig. Sequence alignments of intergenic spacers in *Formica* mitochondrial genomes.** (A) The intergenic spacer between *trnI* and *trnQ*. (B) The intergenic spacer between *trnH* and *nad4*. (C) The intergenic spacer between *trnQ* and *nad2*. (D) The intergenic spacer between *trnC* and *trnY*. (E) The intergenic spacer between *trnY* and *cox1*. (F) The intergenic spacer between *cox1* and *trnL2*. (G) The intergenic spacer between *cox2* and *trnK*. (H) The intergenic spacer between *atp6* and *cox3*. (I) The intergenic spacer between *atp6* and *cox3*, except in *F. neogagates*. (J) The intergenic spacer between *cox3* and *trnG*. (K) The intergenic spacer between *trnR* and *trnN*. (L) The intergenic spacer between *trnN* and *trnS1*. (M) The intergenic spacer between *trnS1* and *trnE*. (N) The intergenic spacer between *trnF* and *nad5*. (O) The intergenic spacer between *trnF* and *nad5* in *F. glauca*, *F. sinae*, *Formica sp.DM659*, *Formica* sp. DM658, and *Formica* sp.DM656. (P) The intergenic spacer between *trnF* and *nad5* in *F. candida*, *F. fusca*, *F. moki*, *F. podzolica*, *F. rufa*, and *F. selysi*. (Q) The intergenic spacer between *nad4* and *nad4L*. (R) The intergenic spacer between *cob* and *trnS2*.
(TIF)

**S3 Fig. Phylogenetic relationships among 16 Formicinae species in four genera based on two datasets (P123 and P123AA) and three analytical methods (Bayesian inference [BI], neighbor-joining method [NJ], and maximum likelihood [MJ]).**
(TIF)

**S1 Table. Sampling information of the two *Formica* species that were newly sequenced in this study.**
(DOCX)

**S2 Table. Characteristics of mitogenomes of 16 Formicidae species analyzed in this study.** Species that were newly sequenced in this study are labelled with an asterisk.
(DOCX)

**S3 Table. Saturation test for each of the 13 protein-coding genes (PCGs) sequences, concatenated sequences of 13 PCGs and 2 rRNAs, and three codon positions of 13 PCGs as**

implemented in DAMBE.
(DOCX)

**S4 Table. The best partitioning schemes and substitution models selected by IQ-TREE for the two datasets.**
(DOCX)

**S5 Table. Codon usage for the 13 mitochondrial protein-coding genes in 12 *Formica* mitochondrial genomes.**
(XLSX)

## Author Contributions

**Conceptualization:** Ming-Long Yuan.

**Formal analysis:** Min Liu, Shi-Yun Hu, Min Li, Hao Sun.

**Funding acquisition:** Ming-Long Yuan.

**Investigation:** Min Liu, Shi-Yun Hu.

**Methodology:** Min Liu, Min Li, Hao Sun.

**Software:** Shi-Yun Hu.

**Supervision:** Ming-Long Yuan.

**Visualization:** Min Liu, Shi-Yun Hu, Min Li, Hao Sun.

**Writing – original draft:** Min Liu, Shi-Yun Hu.

**Writing – review & editing:** Min Li, Hao Sun, Ming-Long Yuan.

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
