## [Decision Letter · Decision Letter 0]

6 Feb 2024

PONE-D-23-17435Comparative mitogenomic analysis provides evolutionary insights into Formica (Hymenoptera: Formicidae)PLOS ONE

Dear Dr. Yuan,

Thank you for submitting your manuscript to PLOS ONE. After careful consideration, we feel that it has merit but does not fully meet PLOS ONE’s publication criteria as it currently stands. Therefore, we invite you to submit a revised version of the manuscript that addresses the points raised during the review process.

Both reviewers have concerns on the presentation and writing, which should be improved in the new version.==============================

We look forward to receiving your revised manuscript.

Kind regards,

Yanbin Yin

Academic Editor

PLOS ONE

Journal Requirements:

"This study was funded by the Second Tibetan Plateau Scientific Expedition and Research (STEP) Program (2019QZKK0302), and the Key Project of Science and Technology Department of Xinjiang Autonomous Region, China [2020E0213, 2016A03006].”

“This study was funded by the Second Tibetan Plateau Scientific Expedition and Research (STEP) Program (2019QZKK0302), and the Key Project of Science and Technology Department of Xinjiang Autonomous Region, China [2020E0213, 2016A03006]. The funders had no role in study design, data collection and analysis, decision to publish, or preparation of the manuscript.”

Reviewers' comments:

Reviewer's Responses to Questions

**Comments to the Author**

1. Is the manuscript technically sound, and do the data support the conclusions?

Reviewer #1: Yes

Reviewer #2: Yes

2. Has the statistical analysis been performed appropriately and rigorously? 

Reviewer #1: Yes

Reviewer #2: Yes

3. Have the authors made all data underlying the findings in their manuscript fully available?

Reviewer #1: Yes

Reviewer #2: Yes

4. Is the manuscript presented in an intelligible fashion and written in standard English?

Reviewer #1: Yes

Reviewer #2: Yes

5. Review Comments to the Author

Reviewer #1: The manuscript investigated two mt-genomes of the genus Formica and resconstructed the phylogenic relationship within the genus using those datasets. The results will be a potential reference for the understanding of insects mt-genome and their applification for the phlogeny within genus.

Reviewer #2: The authors sequenced two Formica species and assembled the mitogenomes, then the authors did some analysis, such as coding genes, tRNAs, codon usage. I have some comments as follows:

1. I suggest the authors marked the ‘Line numbers’ throughout the manuscript, it would be easier to track the comments.

2. In the section ‘3.2. Nucleotide composition and codon usage’, keep the number ‘3274’ and ‘3712’ (be 3,712) the same style with previous content, please check with all the manuscript.

3. In the section ‘3.3. Intergenic spacers’, I suggest rephrase the intergenic spacers (IGSs), especially you don’t need to list all the 14 spacers in details, please emphasize most important results.

4. In the section ‘3.6. Mitochondrial phylogeny of Formica’, there is a typo error ‘group 2 including five species, with.’

5. In the section ‘4.5. Formica Phylogeny and intergenic spacers’, the authors said ‘unstable phylogenetic relationships of Formica’, which phylogeny gives most confident their relationships? What’s the explanation?

6. Figure 9, I suggest the authors label all the nodes with bootstrap values.

6. PLOS authors have the option to publish the peer review history of their article (what does this mean?). If published, this will include your full peer review and any attached files.

Reviewer #1: No

Reviewer #2: No

---

## [Author Response · Author response to Decision Letter 0]

21 Mar 2024

For Reviewer #1

The manuscript investigated two mt-genomes of the genus Formica and resconstructed the phylogenic relationship within the genus using those datasets. The results will be a potential reference for the understanding of insects mt-genome and their applification for the phlogeny within genus.

Response: We thank the reviewer’s positive comments on the manuscript.

For Reviewer #2 

1. I suggest the authors marked the ‘Line numbers’ throughout the manuscript, it would be easier to track the comments.

Response: We have provided “Line numbers” throughout the revised manuscript, as suggested.

2. In the section ‘3.2. Nucleotide composition and codon usage’, keep the number ‘3274’ and ‘3712’ (be 3,712) the same style with previous content, please check with all the manuscript.

Response: We have revised the manuscript, as suggested.

3. In the section ‘3.3. Intergenic spacers’, I suggest rephrase the intergenic spacers (IGSs), especially you don’t need to list all the 14 spacers in details, please emphasize most important results.

Response: We have rephrased the intergenic spacers, by deleting some unnecessary descriptions of 14 IGSs to emphasize the most important results, as suggested.

4. In the section ‘3.6. Mitochondrial phylogeny of Formica’, there is a typo error ‘group 2 including five species, with.’

Response: We have changing “group 2 including five species, with” into “group 2 including five species”, as suggested.

5. In the section ‘4.5. Formica Phylogeny and intergenic spacers’, the authors said ‘unstable phylogenetic relationships of Formica’, which phylogeny gives most confident their relationships? What’s the explanation?

Response: We thank the reviewer’s invaluable comments. The Formica phylogeny (Figure 9A) obtained from the P123 dataset may be most confident, as reported by Goropashnaya et al. (PLoS One, 2012). We discussed the unstable phylogenetic relationships of Formica in the Discussion section.

6. Figure 9, I suggest the authors label all the nodes with bootstrap values.

Response: We thank the reviewer’s invaluable comments. Given the six Formica phylogenies have almost consistent phylogenetic relationships and most support values were more than 90 and 0.9, we integrated all phylogenetic results (Figure 3A) into a common phylogenetic backbone and used black dots to denote support values greater than 90 (RA x ML) and 0.9 (MrBayes and NJ), which did not affect the understanding of the phylogenies. We also provided all phylogenetic trees with specific support values for each node in Figure 3A.

---

## [Editor Report · Decision Letter 1]

3 Apr 2024

Comparative mitogenomic analysis provides evolutionary insights into Formica (Hymenoptera: Formicidae)

PONE-D-23-17435R1

Dear Dr. Yuan,

We’re pleased to inform you that your manuscript has been judged scientifically suitable for publication and will be formally accepted for publication once it meets all outstanding technical requirements.

Kind regards,

Yanbin Yin

Academic Editor

PLOS ONE
---

## [Editor Report · Acceptance letter]

30 May 2024

PONE-D-23-17435R1 

PLOS ONE

Dear Dr. Yuan, 

I'm pleased to inform you that your manuscript has been deemed suitable for publication in PLOS ONE. Congratulations! Your manuscript is now being handed over to our production team.

Kind regards, 

on behalf of

Dr. Yanbin Yin 

Academic Editor

PLOS ONE